# The Italian Validation of the Zimbardo Time Perspective Inventory and Its Comparison with Three Time Perspective Inventories

**DOI:** 10.3390/ijerph20032590

**Published:** 2023-01-31

**Authors:** Monica Martoni, Marco Fabbri, Paolo Maria Russo

**Affiliations:** 1Department of Medical and Surgical Sciences, Alma Mater Studiorum, University of Bologna, 40126 Bologna, Italy; 2Department of Psychology, University of Campania “Luigi Vanvitelli”, 81100 Caserta, Italy

**Keywords:** time perspective, Zimbardo time perspective inventory, psychometric properties, confirmatory factor analysis, exploratory structural equation modeling, reliability, high frequency items, ZTPI-16, Italian culture

## Abstract

The Zimbardo time perspective inventory (ZTPI) is the most well-known and widely used measure of time perspective. However, the assessment of the psychometric properties of the ZTPI reveals several problems, and various short versions have been proposed to overcome these problems. In a large Italian sample (N = 2295; 1326 women; age range 18–74 years), the present study aimed to test a short version of the ZTPI (ZTPI-16) defined by high frequency items (i.e., “good” items), reviewing the items composition of previous alternative short versions of the scale. Beyond the assessment of the factorial structure of this new short ZTPI, we compared the ZTPI-16 to the original ZTPI (ZTPI-56) and another already validated version of the ZTPI in the Italian context, such as Zimbardo’s Stanford time perspective inventory (ZTPI-22), the short version of the ZTPI (ZTPI-30), and the ZTPI-36 proposed analyzing the data from 24 countries. The results confirmed the psychometric problems of the ZTPI-56, whereas the ZTPI-16 reported adequate structural validity and reliability. Moderate-to-strong correlations between same temporal subscales in different ZTPI versions were also found. These data suggest that the review of the “good” items is a new direction in the development of ZTPI versions with good psychometric properties and comparable data among cultures.

## 1. Introduction

Time perspective (TP) defines a process by which individuals automatically categorize the flow of their personal experiences into psychological time frames of past, present, and future [1]. These temporal frames influence several human behaviors, such as risk taking, health-promoting behavior, quality of life, and sleep quality [2,3,4,5]. In the literature, TP can be related to individual differences [6,7,8,9,10], socioeconomic status [11,12,13], cultural effects [14,15,16,17,18], and different personality traits [19,20]. However, the conceptualization and measurement of the construct have been problematic. Thus, in the time perspective literature, the measurement issue is still an open question.

One of the measures to capture TP is the Circles Test [21], which requires participants to draw circles to represent their past, present, and future and then to arrange the circles in any way they want. The relatedness and the relative size of the circles reflect the participant’s time orientation [22,23]. Another test is the Time Structure Questionnaire (TSQ), which measures how individuals perceive their utilization of time as structured and goal-directed [24]. In other words, the TSQ is designed to measure one’s attitude toward time [25]. However, the most widely used questionnaire for the time perspective is the Zimbardo time perspective inventory (ZTPI) [1], with a solid theoretical background and which covers different dimensions of time perspective [1,18]. Indeed, the ZTPI encompasses five subscales: past positive (PP) involves a positive, warm, and nostalgic view of the past; past negative (PN) captures a negative and aversive view of the past; present hedonistic (PH) reflects the attitude toward the present, involving immediate pleasure seeking with little consideration of future consequences; present fatalistic (PF) reflects a hopeless and helpless view of the present, where present behavior is considered as irrelevant to future consequence; future (F) reflects a broad orientation toward the future, involving optimism and striving for future goals and rewards.

To the best of our knowledge, the ZTPI has been translated into several languages, and different adaptations and/or versions have been proposed [26,27]. Various studies have investigated the psychometric properties of the ZTPI, reporting adequate internal reliability (from 0.63 to 0.84, in different countries, such as Brazil [28], China [29] Estonia [30], Greece [31], Japan [32], Latvia and Russia [33], Lithuania [34], Mexico [35], Norway [36], Poland [37], Nigeria [37], United States [37], Sweden [38], Turkey [39], and the United Kingdom and Russia [40]. When the structural validity has been investigated, the studies have reported mixed results. On the one hand, some studies replicated the five-factor structure of the ZTPI, using exploratory factor analysis (EFA) [26,31,41,42,43] and/or confirmatory factor analysis (CFA) [29,44,45]. On the other hand, several studies reported poor structural validity of the ZTPI, and different items did not fit with the original time perspective subscale [26,28,29,37,44,46,47,48,49]. In addition, different models with one-, three-, four-, six-, or seven-subscale solutions have been proposed in different contexts [31,36,37,38,40,50]. This volatile factorial validity across studies and cultures has determined the development of a shortened version of the ZTPI, given that the original version of the ZTPI, with 56 items, has showed relatively poor EFA and CFA model fits. Indeed, to the best of our knowledge, in the literature there are shorter versions of the ZTPI in Chinese [29,51], Czech and Slovak [52], Indian [53], Hungarian [45], Hebrew [54], Poland [37], Italian [50,55,56], Slovenian, American and British [47], Norwegian [36], and Australia [48], see also [26] contexts, proposing scales with 15 or 20 items, 15 (or 18) items, 16 items, 17 items, 20 items, 20 items, 22, 25 or 30 items, 34 items, and 36 items, respectively. All these versions have reported acceptable psychometric properties suggesting that a shorter version of the original ZTPI could be considered (and used more) as a way to address and resolve the measurement issues reported in the time perspective field with the most widely used questionnaire (ZTPI).

However, these shorter versions have often not been reproduced in other samples. In line with this limit, Temple et al. [57] questioned these strategies to strengthen the reliability and validity of the ZTPI solely through the selection of specific items, proposing a shortened version of the scale, generally depending on the cultural context of the study. Specifically, the authors recommended a more theoretical approach to assess the time perspective with a tool which transcends cultural differences. In line with these recommendations, Temple et al. [57] demonstrated that a data-driven approach limited the generalizability of these shortened versions, given that they demonstrated in different samples from different countries (i.e., Australia, Britain, Slovenia, and the United States) that the 15-, 17- and 20-item versions of the ZTPI reported low psychometric properties. In a similar way, Perry et al. [47] revealed unsatisfactory internal consistency and factorial validity for the 25-item version in Slovenian, American, and British adolescents. Moreover, McKay et al. [48], for the sample of adolescents (from the United Kingdom and the United States) and adults (from Australia), found a good internal consistency of the ZTPI with 36 items provided by [26], but the structural validity reported poor fit indexes and numerous problematic items, reinforcing a more theoretical approach to modify the ZTPI; for example, eliminating items which do not assess time perspective [46]. When Worrell et al. [58] adopted a theory-driven approach to enhance the psychometric validity of the ZTPI, including only items with a specific temporal context, a new 25-item version was provided and tested in a large cross-cultural study. The authors reported acceptable cross-cultural indexes, suggesting that the proposed version with items referring to only temporal aspects could resolve the psychometric problems of the ZTPI. However, this 25-item version reported problematic indexes for PF and PH subscales, limiting the possibility that this short version of the ZTPI could be a “final answer” of the measurement issue.

Although Worrell et al. [58] proposed the need to rewrite the items of the ZTPI with more focus on the time-based reference, Peng et al. [27] have recently proposed a new strategy to address conceptual and measurement concerns regarding the ZTPI. The authors systematically reviewed the findings of previous short versions of the ZTPI and identified the “good” and “bad” items for increasing the psychometric properties of the ZTPI. According to the authors, the systematic review can provide a clear method to summarize the findings of previous studies, for e.g., [59], by providing a new short version of the ZTPI, which should be reliable, valid, and culturally independent. Peng et al. [27] calculated the frequency of each item in all short versions of the ZTPI, reported in the literature and considered in their systematic review. At this point, the authors obtained high, medium, and low frequent forms of the ZTPI based on the frequency of items (e.g., a high frequent form was composed of high frequency items). In order to assess a possible five-factor structure for each short form, Peng et al. [27] proposed that each short form was composed of 16 items (see Table 1, page 4 in [27]). The authors [27] reported that the high frequent form had the best fit index, followed by the medium frequent form, and then the low frequent form, whereas the original version of the ZTPI reported poor fit indexes. In addition, the high frequent form reported good internal consistency, although, in this case, the 56-item version of the ZTPI obtained better reliability values. Thus, the authors concluded that the items with a higher frequency were more robust in measuring the time perspective and recommended to adopt this approach in further works to exclude any potential cultural influence in the psychometric properties of short versions of the ZTPI. This study was a possible attempt to confirm the findings reported by Peng et al. [27] in an Italian context. Specifically, we assessed the reliability and the structural validity of short ZTPIs composed of high frequency items in a large Italian sample. Taking into account the popularity and the use of the original ZTPI in the time perspective literature [1,26,27,58], as well as its associations with a range of human behaviors [2,3,4,5], we considered the present research as an attempt to address the measurement problem of the ZTPI, testing a promising 16-item solution.

In line with this recommendation, the aim of the present study was to propose and test a new short version of the ZTPI, comparing the short form provided by Peng et al. [27] (i.e., ZTPI-16) with an original, 56-item ZTPI (in the present study we used the term ZTPI-56) in an Italian context, in a similar way to the procedure adopted by [27] and [52]. In Italian culture, we found three different versions of the ZTPI: D’Alessio et al. [50] evaluated the Zimbardo’s Stanford time perspective inventory (STPI, but in the present study we used the term ZTPI-22), short form, with PH, PF, and F subscales in a 22-item form; Laghi et al. [55] provided a 25-item version of the ZTPI with all five time perspectives in adolescents only (but see [47] for contrasting results); and Molinari et al. [56] assessed a short version of the ZTPI (S-ZTPI, but in the present study we used the term ZTPI-30), a 30-item version with six subscales splitting the F time perspective into future positive (FP) and future negative (FN) [60]. In addition, Fabbri et al. [5] used a translated version of the original ZTPI-56, only providing the reliability for each subscale, whereas no further assessment of the psychometric properties was addressed in that study. Although the ZTPI-22 and ZTPI-30 reported good psychometric properties, it is still missing an agreement for the (right) tool to be used for assessing the time perspective in Italian context. Thus, the assessment of the structural validity and reliability of the proposed ZTPI-16 could provide evidence for the utility of this short version of the ZTPI, not only for the Italian context but also in other contexts, improving the concept and the measurement of time perspective. In line with Peng et al. [27], we expected to obtain poor model fits and high reliability for the original ZTPI-56, whereas good structural validity and adequate reliability for the ZTPI-16 composed of high frequency items [27] were also expected. In order to further test the psychometric properties of this 16-item version of the ZTPI, in the Italian context, we correlated it with ZTPI-22, ZTPI-30, and ZTPI-36, the latter provided by Sircova et al. [26] who assessed the psychometric properties of the ZTPI in samples of convenience from 24 countries. The choice of the ZTPI-22, ZTPI-30, and ZTPI-36 was based on the fact that these versions were validated in samples similar to that recruited in the present study (e.g., age range 16–89 years [50]), they were validated in the Italian context or in a large cross-cultural study, provided good psychometric properties, and could be used for testing a concurrent validity (for similar procedure see [52]).

## 2. Materials and Methods

### 2.1. Participants and Procedure

A sample of 2295 volunteers participated in the survey. Participants were unpaid, anonymous, and could be withdrawn at any time. Of the participants, 1326 were women and 969 were men. The mean age was 30.36 years (SD = 12.99 years; age range 18–74 years), and there was no age difference between the women (M = 30.01 years; SD = 12.74 years) and men (M = 30.84 years; SD = 13.34 years), with *t*(2293) = 1.52, *p* = 0.13, *Cohen’s d* = 0.06). As noted by [27], it is a challenge to determine the requirements of the sample size for CFA (see page 3 of [27] for relative references about this topic), and, thus, we compared our sample size to that reported in the literature. Sircova et al. [26] included 26 samples for their cultural comparison of the ZTPI and covered studies from 2004 to 2012 with sample sizes ranging from 180 to 1.269. From 2013 to 2021, the numerosity of the sample size in single studies ranged from 187 [53] to 2.062 [52], while that in the multicenter (and/or cross-cultural) studies ranged from 1.150 [37] to 3.306 [58]. In addition, in the Italian context, D’Alessio et al. [50] recruited 1.507 individuals, Laghi et al. [55] included 435 adolescents, and Molinari et al. [56] assessed 435 participants. Thus, our 2.295 volunteers seemed to avoid the possible influence of an insufficient sample size on conclusions.

The majority (N = 1279; 55.73%) of participants were recruited from university courses, whereas the other participants were recruited through flyers and social media posts (for similar procedure see [28,29,36,39,44,45,48,49,55,56,57,58]). Thus, participants were selected on the basis of their willingness to participate, and a convenience sample was recruited through opportunistic and snowball sampling. About half of the sample (N = 1124; mean age of 31.07 ± 13.32 years; age range 18–69; 664 women and 460 men) were also administered the ZTPI-22 (N = 708), the ZTPI-30 (N = 216), or the ZTPI-36 (N = 200), altogether with the ZTPI-56, whereas the remaining half of the sample received the original ZTPI only. The different rate of participants in administering an additional version of the ZTPI was based on the higher number of citations received by ZTPI-22 (N = 160 in Scopus vs. N = 8 and N = 114 in Scopus for the ZTPI-30 and the ZTPI-36, respectively). Given that we proposed a 16-item version, we decided that it was more useful to compare our ZTPI-16 with another very short scale (ZTPI-22) with respect to ZTPI versions with more items. All participants compiled questionnaires individually after receiving a brief explanation of the study. When two versions of the ZTPI were provided to participants, the order of versions was counterbalanced. The study protocol was approved by the Ethical Committee of Department of Psychology (protocol number: 1/2018 approved 6 March 2018), at University of Campania “Luigi Vanvitelli”, and all participants provided informed consent prior to filling in the questionnaire.

### 2.2. Zimbardo Time Perspective Inventory (ZTPI-56)

To evaluate the TP, we administered the Italian version of the ZTPI [1], adopted by Fabbri et al. [5]. The Italian version of the ZTPI contained 56 items rated on a 5-point Likert scale (1 = very uncharacteristic of me; 5 = very characteristic of me). The ZTPI identifies five TP dimensions: past negative (PN: “*I often think about the bad things that have happened to me in the past*”), past positive (PP: “*I enjoy stories about how things used to be in the good old times*”), present hedonistic (PH: “*I take risks to put excitement in my life*”), present fatalistic (PF: “*Since whatever will be will be, it doesn’t really matter what I do*”) and future (F: “*Meeting tomorrow’s deadline and doing other necessary work comes before tonight’s play*”).

### 2.3. Short Version of the Zimbardo Time Perspective Inventory (S-ZTPI-30 or ZTPI-30)

The S-ZTPI or ZTPI-30 [60], consisting of thirty items and six subscales composed of five items each, was adopted in Italian culture by Molinari et al. [56]. The subscales are PN (“*Painful past experiences keep being replayed in my life*”), PP (“*Happy memories of good times spring readily to mind*”), PH (“*I find myself getting swept up in the excitement of the moment*”), PF (“*Fate determines much in my life*”), future positive (FP: “*I complete projects on time by making steady progress*”) and future negative (FN: “*Usually, I don’t know how I will be able to fulfil my goals in life*”). For the ZTPI-56, responses were given on a 5-point Likert scale (1 = very uncharacteristic of me; 5 = very characteristic of me). Molinari et al. [56] reported the following Cronbach’s alpha (α) for each subscale: α = 0.73, α = 0.70, α = 0.66, α = 0.65, α = 0.72, and α = 0.65 for PN, PP, PH, PF, FP, and FN, respectively. In the present research the internal reliability of each subscale was equal to α = 0.76, α = 0.78, α = 0.70, α = 0.62, α = 0.80, and α = 0.64 for PN, PP, PH, PF, FP, and FN, respectively.

### 2.4. Stanford Time Perspective Inventory—Short Form (STPI-22 or ZTPI-22)

The STPI or ZTPI-22 [50] contains 22 items rated on a 5-point Likert scale (1 = very uncharacteristic of me; 5 = very characteristic of me) and identifies 3 time perspective subscales: PH (8 items with 2 reversed items; “*I believe that getting together with one’s friends to party is one of life’s important pleasures*”), PF (5 items; “*It doesn’t make sense to worry about the future since there is nothing to do about it anyway*”), and F (9 items; “*I believe that a person’s day should be planned ahead each morning*”). D’Alessio et al. [50] provided the following Cronbach’s alpha: α = 0.54, α = 0.49, and α = 0.67, for PH, PF, and F respectively. In the present research the internal reliability of each subscale was equal to α = 0.48, α = 0.62, and α = 0.77 for PH, PF, and F, respectively.

### 2.5. Zimbardo Time Perspective Inventory—36 Items (ZTPI-36)

The ZTPI-36 [26] is comprised of 36 items assigned to 5 factors: PN (items 2, 11, 15, 16, 21, 23, and 32 from the original ZTPI-56; e.g., “*I often think of what I should have done differently in my life*”), PP (items 1, 3, 7, 17, 31, and 35 from the original ZTPI-56; e.g., “*On balance, there is much more good to recall than bad in my past*”), PH (items 4, 8, 9, 10, 13, 19, 27, 28, 34, and 36 from the original ZTPI-56; e.g., “*I try to live my life as fully as possible one day at a time*”), PF (items 14, 20, 22, 24, 25, and 30 from the original ZTPI-56; e.g., “*My life is controlled by forces I cannot influence*”), and F (items 5, 6, 12, 18, 26, 29, and 33 from the original ZTPI-56; e.g., “*When I want to achieve something, I set goals and consider specific means for reaching those goals*”). Participants responded to questions using a 5-point Likert scale (1 = very uncharacteristic of me; 5 = very characteristic of me). The items were taken from the Italian translation of the original ZTPI-56 provided by Fabbri et al. [5]. Sircova et al. [26] reported the following Cronbach’s alpha (α) for each time perspective: α = 0.77, α = 0.66, α = 0.69, α = 0.60, and α = 0.64, for PN, PP, PH, PF, and F, respectively. In our study, the reliability of each subscale was equal to α = 0.70, α = 0.73, α = 0.71, α = 0.52, and α = 0.80 for PN, PP, PH, PF. and F respectively.

### 2.6. Data Analysis

First, we assessed the psychometric properties of the original ZTPI-56 version, and thus means, standard deviation, median, skewness, and kurtosis were calculated. In addition, the inter-item correlations were calculated. Cronbach’s alpha (α) was used to assess the internal consistency in the scale and the correlations between scales were also calculated. Second, the factorial structure of the ZTPI-56 was assessed through exploratory structural equation modeling (ESEM) by means of Mplus software [61]. The ESEM method performs better than CFA and could be considered a preferred approach to examine model fits in multidimensional measures [62]. The ESEM assessed the proposed 5-factor structure [1] and the solution was generated on the basis of the robust maximum likelihood estimation (MLR). In addition, the ESAM analyses were conducted using a multiple-groups procedure, that is, the total sample was divided into two separate samples: the first one for the calibration, and the second sample for the cross-validation. Each participant was randomly assigned to one of these two groups. This approach is the most commonly used for the measurement invariance testing of attitudinal measures. The multigroup method assumes the equality of model parameters in both groups (full measurement invariance) and allows for the evaluation of three hierarchical levels of measurement invariance through the comparison of different models with increasing constraints: (a) configural invariance, (b) metric invariance, and (c) scalar invariance [63].

According to Brown [64] and Schreiber et al. [65] several indexes of the goodness of fit were taken into consideration, including the chi-square degree of freedom ratio (*χ^2^*/df), root mean square error of approximation (RMSEA) and its 90% confidence interval (90% CI), and test of close fit (CFit), the comparative fit index (CFI), the Tucker–Lewis index (TLI) and the standardized root mean square residuals (SRMR) were also calculated. In line with the suggestions provided by Hu and Bentler [66], the acceptable model fit was defined using the following criteria: RMSEA (≤0.06), CFI (≥0.90), TLI (≥0.95), and SRMR (≤0.08). However, Perry et al. [67] suggested that strict adherence to these cut-off values often leads to erroneous results, given that factor loadings in social sciences are typically lower [68]. The same statistical analyses, in terms of reliability and factor structure, were performed for the proposed model (ZTPI-16) of the ZTPI, comprising 16 items and 5 TP subscales. This ZTPI-16 was composed of high frequency items, according to Peng et al. [27]. Specifically, the items (derived from the original ZTPI-56) of each subscale of our model were: 16, 34, and 50 for PN; 2, 7, and 20 for PP; 26, 31, and 42 for PH; 37, 38, and 39 for PF; 13, 21, 40, and 45 for F (see Appendix A).

Finally, we performed correlations analyses between corresponding TP subscales of the ZTPI-56 (i.e., PN-56, PP-56, PH-56, PP-56, and F-56) and those of ZTPI-16 (i.e., PN-16, PP-16, PH-16, PF-16, and F16), as well as with TP dimensions of ZTPI-30 (i.e., PN-30, PP-30, PH-30, PF-30, FP-30, and FN-30), ZTPI-22 (i.e., PH-22, PF-22, and F-22), and ZTPI-36 (i.e., PN-36, PP-36, PH-36, PF-36, and F-36). Also, we performed a set of analyses with demographic information.

## 3. Results

The descriptive data of the ZTPI-56 are reported in Table 1. The internal consistency of the ZTPI-56 assessed using Cronbach’s alpha for each subscale was: α = 0.81, α = 0.75, α = 0.79, α = 0.74, and α = 0.74 for PN-56, PP-56, PH-56, PF-56, and F-56, respectively. The correlations between time perspectives ranged from −0.21 (PF-56 × F-56) to +0.40 (PN-56 × PF-56) and were significant, except for the correlation between PP-56 and PN-56 (*r* = −0.03, *p* = 0.18), between PP-56 and PF-56 (*r* = +0.04, *p* = 0.08), and between PN-56 and F (*r* = +0.03, *p* = 0.15). Thus, we found negative correlations between F-56 and PH-56 (*r* = −0.06, *p* < 0.005) and PF-56 (*p* < 0.0001). On the other side, significant associations (all with *p* < 0.0001) were found between PP-56 and PH-56 (*r* = +0.18), PP-56 and F-56 (*r* = +0.24), PN-56 and PH-56 (*r* = +0.20), PN-56 and PF-56 (*r* = +0.40), and between PH-56 and PF-56 (*r* = +0.30). The structural validity of the ZTPI-56, assessed by conducting ESEM, demonstrated low fit indexes: *χ^2^* (1475) = 7974.71, *p* < 0.0001, RMSEA = 0.06, 90% CI = 0.058−0.060, CFI = 0.60, TLI = 0.58, SRMR = 0.091. Altogether these indexes indicated inadequate model fit and cross-validation procedure were not performed.

The descriptive data of the ZTPI-16 are reported in Table 1, while the factorial structure of ZTPI-16 is shown in Figure 1. The Cronbach’s alpha for each subscale was equal to α = 0.75, α = 0.67, α = 0.60, α = 0.70, and α = 0.66, for PN, PP, PH, PF, and F, respectively, confirming previous results of a higher reliability for the longer version than the shorter version of the ZTPI [27]. In order to estimate the measurement reliability of the total score, the coefficient omega (ω) for each subscale of the ZTPI-16 was used [69]. In the present study, we found ω = 0.76, ω = 67, ω = 0.66, ω = 0.70, and ω = 0.65 for PN-16, PP-16, PH-16, PF-16, and F-16, respectively. Thus, internal consistency estimates were acceptable for all TP subscales, especially when, in line with [58], a value of 0.60 was employed as acceptable. The correlations between ESEM time perspective factors were calculated and significant, positive correlations were found between PH and PN (*r* = 0.17, *p* < 0.001), PH and PF (*r* = 0.19, *p* < 0.001), PF-16 and PN-16 (*r* = 0.32, *p* = 0.28), and PP-16 and F-16 (*r* = 0.38, *p* < 0.001). Furthermore, we observed negative significant associations between F-16 and PF-16 ((*r* = −0.15, *p* < 0.001), as well as between PH-16 and F-16 (*r* = −0.22, *p* < 0.001). The ESEM model showed marginally adequate fit indexes for a five-factor structure: RMSEA = 0.061, 90% CI = 0.056–0.066, CFI = 0.87, TLI = 0.83, and SRMR = 0.060. Thus, the ZTPI-16 seemed to yield more adequate fit indexes than the original ZTPI-56. The results of the ESEM for ZTPI-16 were cross-validated with a multigroup approach. Multigroup analyses evidenced that ZTPI-16 shows satisfactory metrical (metric against configural *χ^2^* difference = 3.93, df = 11, *p* = 0.97) and scalar invariance (scalar against metric *χ^2^* difference = 1 4.71, df = 11, *p* = 0.19).

Table 2 shows the Pearson *r* correlations between the ZTPI-16 and ZTPI-56, ZTPI-36, ZTPI-30, or ZTPI-22. Observing Table 2, we observed significant moderate-to-strong (from +0.40 to +0.85, except weak +0.25 for the association between PH-16 and PH-36) positive correlations between the five time perspectives identified via the proposed ZTPI-16 and the corresponding time perspective identified in every alternative ZTPI used in the present study (see values in bold in Table 2). In addition, we confirmed the associations between specific time perspectives in line with what has been observed when correlations between subscales of a specific version of the ZTPI was performed. For example, the PN-16 correlated negatively with PP-56 and positively with PF-56, confirming previous results [29,37,45,52,54]. Interestingly, we observed positive correlations between PP-16 and FP-30, as well as positive correlations between FN-30 and PN-16, PH-16, PF-16, as reported by [38,56,60]. Altogether, these results suggested a concurrent validity of the ZTPI-16. In other words, these correlations seemed to suggest that the ZTPI-16 measured TP construct in similar way to other validated ZTPI versions.

When we correlated the age with each time perspective score, we found significant correlations between age and PH-16 (*r* = −0.26, *p* < 0.0001), PF-16 (*r* = +0.26, *p* < 0.0001), and F-16 (*r* = −0.04, *p* < 0.05). In addition, we found gender differences for all time perspectives. Specifically, women were more PP-16 (M = 3.64; SD = 0.80)- and PN-16 (M = 2.69; SD = 1.02)-oriented than men (3.48 ± 0.91, and 2.56 ± 1.02, respectively) with *t*(2293) =−4.51, *p* < 0.0001, *Cohen’s d* = 0.19, and *t*(2293) = −2.94, *p* < 0.005, *Cohen’s d* = 0.13. In addition, women reported higher PF-16 (M = 2.41; SD = 0.90) and F-16 (M = 3.61; SD = 0.69) scores than men (2.31 ± 0.89, and 3.50 ± 0.76, respectively), with *t*(2293) = −2.67, *p* < 0.05, *Cohen’s d* = 0.11 and *t*(2293) = −3.55, *p* < 0.0001, *Cohen’s d* = 0.15. By contrast, men were more PH-16 (M = 2.91; SD = 0.84)-oriented than women (M = 2.63; SD = 0.81), with *t*(2293) = +8.15, *p* < 0.0001, *Cohen’s d* = 0.34.

## 4. Discussion

The present study aimed to propose and test a short version of the ZTPI-56—that is the ZTPI-16 [27] in the Italian context. This aim was grounded on the possibility to test the approach adopted by Peng et al. [27] in addressing the measurement issue for the ZTPI, which is the most widely used tool in the time perspective literature. The results showed that the version of the ZTPI composed of the items with a higher frequency as proposed by Peng et al. [27] was a more adequate scale for measuring the time perspective than the original version. Indeed, the present study added further evidence that, even for the Italian context, the ZTPI-56 was inadequate in its structural validity, in line with previous studies [26,28,29,37,44,46,47,48,49]. This result was found by recruiting university students, adults, and old adults, while previous studies were conducted on adolescents [44,47,55,56] or adults [28,30,31,36,38,39,53], confirming that the original ZTPI-56 is problematic probably because some items measure other constructs rather than the time perspective [39,58]. Indeed, the CFA indexes found in the present study for the ZTPI-56 (RMSEA = 0.06, 90% CI = 0.058−0.060, CFI = 0.60, TLI = 0.58, SRMR = 0.091) not only confirmed those reported by Peng et al. [27] for Chinese children, undergraduates, and old adults (RMSEA = 0.08, CFI = 0.63, TLI = 0.62), but were also in line with CFA indexes reported, for example, by Akirmak [39] (RMSEA = 0.06, 90% CI = 0.056–0.061, TLI = 0.64, SRMR = 0.09), by Carelli et al. [38] (RMSEA = 0.06, CFI = 0.63, RSMR = 0.09), by Milfont et al. [28] (RMSEA = 0.08, CFI = 0.70, TLI = 0.74, SRMR = 0.09), or by Worrell and Mello [44] (90% CI = 0.055–0.059, CFI = 0.64, SRMR = 0.06) for Turkish, Sweden, Brazilian, and American cultures. Although we recognized the recommendations provided by [67] to not be adherent to the cut-off proposed by [66], the fit indices reported by the ZTPI-56 were decisively poor, and the ZTPI-56 probably needs a rephrasing of items to tap TP in a more direct way [39,58]. Taking into account the problematic structural validity of the ZTPI-56, a possible attempt to resolve the psychometric properties of this questionnaire is to shorten it [26,29,36,37,45,47,48,50,51,52,53,54,55,56]. Within the different cross-cultural and shorter version of the ZTPI, in our opinion, the 16-item solution proposed by Peng et al. [27] deserved more attention because it was based on the high frequency items, reviewing all shortened versions of the ZTPI. Although some CFA indexes failed to attain the criterion of good model fit [66] (but see [67] for a different account), the ZTPI-16 seemed to yield more adequate fit indexes than the ZTPI-56. As before, in the present research, we found CFA indexes (CFI = 0.87, TLI = 0.83, RMSEA = 0.061, 90% CI = 0.056–0.066, SRMR = 0.060) similar to those reported by Peng et al. [27] (CFI = 0.97, TLI = 0.96, RMSEA = 0.03). In addition, our indexes were in line with those of both the ZTPI-36 [26] (CFI = 0.86, RMSEA = 0.057, 90% CI = 0.056–0.057, SRMR = 0.06) and the ZTPI-30 [56] (CFI = 0.90, RMSEA = 0.04, SRMR = 0.07), suggesting that the short form with the higher frequency items yielded acceptable CFA results, including only “good” items, representing a possible solution for the psychometric problems of the original ZTPI-56. Furthermore, the multigroup approach confirmed that the ZTPI-16 reported adequate model fit indexes. Our data suggested that the approach provided by Peng et al. [27] could be a possible way to propose an acceptable ZTPI version. Moreover, this approach supported the theoretically driven approach proposed by Worrell et al. [58], given that we confirmed that the “good” items reported a specific temporal content with a clear reference to the past, present, and future. In this way, we attempted to give support, indeed the first of its kind, to the reliability of the method adopted by Peng et al. [27], who systematically reviewed the short versions derived using a data-driven approach for specific samples. The ZTPI-16 could be a reliable tool in addressing (and possibly resolving) the measurement issues in the time perspective literature, given that the similarity of the results between our study in the Italian context and those found by Peng et al. [27] in the Chinese context could indicate that the ZTPI-16 transcends cultural differences, increasing the possibility of generalizing the results obtained for specific cultural samples. Further studies could address this possibility in different cultures, replicating the reliability of this approach, which seems to contribute to the measurement issue in the time perspective literature.

The goodness of the proposed ZTPI-16 was also observed when the internal consistency was assessed. Beyond the replication of similar Cronbach’s alphas between the present study and that performed by Fabbri et al. [5] (here: α was equal to 0.81, 0.75, 0.79, 0.74, and 0.74 for PN-56, PP-56, PH-56, PF-56, and F-56, respectively, vs. Fabbri et al.: α was equal to 0.83, 0.79, 0.85, 0.80, 0.70 for PN, PP, PH, PF, and F, respectively; see also [69]), in line with those reported in other cultural validations of the original ZTPI-56 [26], we also confirmed that the Cronbach’s alpha is influenced by the number of items. This result was consistent with a previous finding that the lower the number of items involved in the questionnaire, the lower the Cronbach’s alpha is [45]. Although shorting the scale reduces the internal consistency, we found acceptable values (α was equal to 0.75, 0.67, 0.60, 0.70, and 0.66, for PN, PP, PH, PF, and F, respectively), in line with Cronbach’s α of other validated ZTPI versions in the Italian context [50,56]. These considerations were also confirmed when omega estimates were calculated, suggesting an acceptable internal consistency in all subscales. Specifically, for PN-16 and PF-16 the coefficient omega was greater than 0.70, and for the remaining three TP subscales the coefficient omega was greater than 0.60, used by [58] as a possible cut-off value of acceptability for internal consistency. In addition, we reported a higher mean IIC within the short version with respect to the long version, suggesting that the selected items measured the different time perspectives. Thus, the ZTPI-16 seemed to provide better fit indexes with a relatively good reliability with respect to what has been reported for the ZTPI-56 [27], another short version of the ZTPI [47,48,57,58], and Italian versions of the ZTPI [50,55,56].

Interestingly, we assessed a possible concurrent validity with other versions of the ZTPI in the Italian context. Table 2 showed that the corresponding subscales (e.g., PP-16 respect to PP-56, PP-36, or PP-30) reported moderate-to-strong correlations, ranging from 0.40 to 0.86, with only a weak correlation between PH-16 and PH-36. However, we recognized a difference in the subscale correlations, given that the *r* values were stronger when the subscales of the ZTPI-16 were correlated with those of the ZTPI-56 (i.e., all *r* values were greater than 0.80, with a *r* = 0.70 for PH) than the correlations found between the ZTPI-16 and ZTPI-30 (correlations ranged from 0.44 to 0.77), ZTPI-36 (correlations ranged from 0.25 to 0.59) or ZTPI-22 (correlations ranged from 0.40 to 0.63). A possible explanation could be related to the fact that the items of the ZTPI-16 were the same items of the ZTPI-56, while slight differences in the item composition of each subscale were observed for other ZTPI versions. Nevertheless, the correlations reported in Table 2 indicated that the short version reported a satisfactory, concurrent validity and that the ZTPI-16 was a useful tool to measure the time perspective in various areas of psychological practice [52]. This aspect was relevant if we considered that our version had fewer corresponding items than those used in other versions of the ZTPI (with the exception of the ZTPI-56) in the Italian context. Moreover, the positive correlations between each subscale of the ZTPI-16 and FN-30, added by Carelli et al. [38] and Molinari et al. [56], suggested that our version could capture the sense of worry and anxiety associated with the future. Related to this point, our data seemed to be in line with the short version in Czech and Slovak cultures for whom the five-factor solution had a slightly better model fit than the six-factor version, due to a strong correlation (*r* = +0.64) between the past negative and future negative in a similar way to our data (*r* = +0.40). Future studies are needed to address the concurrent and predictive validity of our short version, especially with respect to the FN. Also, we not only reported correlations among the subscales of the ZTPI-16 with a similar strength and direction to those reported in other contexts with different samples [26,28,29,37,39,44,45,47,52,53,54], but we also found correlations between the TP subscales of the ZTPI-16 (e.g., PF-16) with other TP scales of ZTPI-22 (e.g., PH-22 and F-22), the ZTPI-30 (e.g., PN-30, PH-30, and FN-30), the ZTPI-36 (e.g., PN-36, PP-36, PH-36, and F-36), and the ZTPI-56 (e.g., PN-56, PH-56, and F-56), generally mirroring similar findings (e.g., positive correlations between PF-16 and PN-30, and PN-36 and PN-56). These findings underlined the usefulness of the 16 items reviewed by Peng et al. [27] for the Italian culture, and further studies on the psychometric properties of this 16-item solution in other cultures are needed. In addition, further studies should deeply address the concurrent validity of the proposed ZTPI-16, examining the relationship of the ZTPI-16 subscale scores with other psychological time-related constructs. Finally, we confirmed a set of gender differences in the time perspective, given that we confirmed a higher score in PP-16, F-16, and PF-16 in women than in men [50,70], while men were more hedonistically oriented than women [70,71]. In a similar way, we partially confirmed the associations between several time perspectives and age [72], although our correlations were weak. We suggest addressing gender- and age-related differences in future studies using our proposed ZTPI-16.

However, the present study was not immune to some limitations. Although we provided similar results reported by Peng et al. [27] in a Chinese sample (i.e., an Asian country), using an Italian sample (i.e., a European country), we cannot rule out that cultural differences exist, leading to a bias in the item selection. For example, the PH-16 in our study was probably the time perspective with low Cronbach’s alpha (and coefficient omega) and low correlation coefficients with other corresponding PH scales (PH-22, PH-30 and PH-36). Thus, further studies may meliorate the item formulation for the PH-16 with a better adaptation to the Italian context. Even if our sample was large and covered a large age range, many participants were recruited in university courses, and further studies, with a more balanced proportion of males and females, should administer the ZTPI-16 to adults and/or old adults in order to address the generalizability of the results obtained with the ZTPI-16 to all populations. Linked to this point, in the present study we decided to include only participants with age ≥18 years and further studies should investigate the application of the ZTPI-16 with adolescents and children. Taking into account that the deviation from the balanced time perspective (DBTP) [19] is the widely used method of calculating how participants can switch effectively among TPs—depending on, for example, task features—in the present study we did not provide any indications to measure this DBTP. We recommend in future studies to indicate how the DBTP could be calculated for the ZTPI-16 and address the association of this index with other psychological constructs.

## 5. Conclusions

To sum up, the present study assessed the psychometric properties of the ZTPI-16 based on the high frequency items reviewed by Peng et al. [27] in an Italian context. In addition, the present study not only assessed the reliability and structural validity of the ZTPI-56 for the first time in Italian culture, but also compared the proposed new version of the ZTPI to the well-validated versions of the scale in Italy. The results confirmed suboptimal model fit indexes for the original version of the ZTPI, although it reported a good reliability, but also confirmed in a European country that the short form of the ZTPI reported adequate structural validity and acceptable internal consistency. The study confirmed that the approach to identify the “good” items for resolving the psychometric properties of the ZTPI is a new direction in promoting the development of research in temporal psychology, probably collocating this approach in a “middle” position between a data-driven approach and a theory-driven approach [57,58]. This approach could be addressed in further investigations in order to obtain a “unique” ZTPI version, transcending cultural differences with the possibility of obtaining comparable data in a different context. This is a first attempt to demonstrate that this short version could resolve the psychometric problems reported by the longer version and be used in the research field of time perspective. The ZTPI-16 seems to be a promising tool for resolving the measurement issue of time perspective, presenting the “good” items of the original ZTPI.

## Figures and Tables

**Figure 1 ijerph-20-02590-f001:**
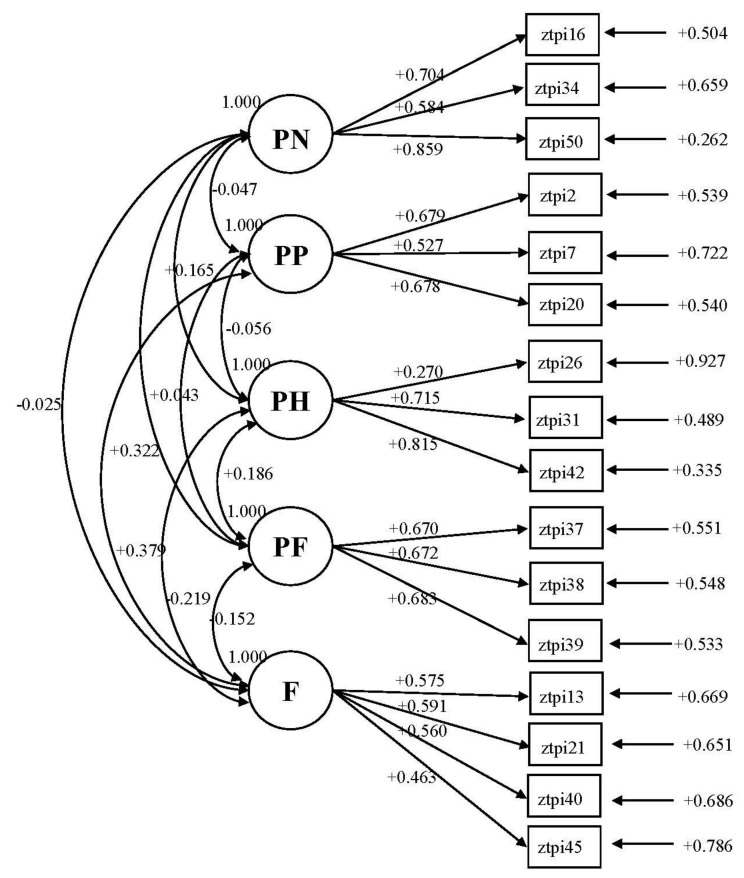
The five-factor model of the ZTPI-16 (N = 2295) with items in each factor structure.

**Table 1 ijerph-20-02590-t001:** The mean (and its SD), median, skewness, kurtosis, and mean inter-item correlations (IIC) for each subscale of the ZTPI-56 and ZTPI-16 are reported.

**ZTPI-56**	**Mean (SD)**	**Median**	**Skewness**	**Kurtosis**	**IIC**
**PN-56**	2.88 (0.74)	2.80	+0.19	−0.44	0.30
**PP-56**	3.53 (0.65)	3.56	−0.51	+0.13	0.25
**PH-56**	3.08 (0.57)	3.07	−0.11	−0.06	0.48
**PF-56**	2.46 (0.65)	2.44	+0.38	+0.0001	0.24
**F-56**	3.50 (0.54)	3.54	−0.35	+0.11	0.18

**ZTPI-16**	**Mean (SD)**	**Median**	**Skewness**	**Kurtosis**	**IIC**
**PN-16**	2.64 (1.02)	2.67	+0.34	−0.59	0.50
**PP-16**	3.57 (0.86)	3.67	−0.58	−0.006	0.40
**PH-16**	2.75 (0.83)	2.67	+0.38	−0.24	0.33
**PF-16**	2.37 (0.90)	2.33	+0.54	−0.19	0.44
**F-16**	3.56 (0.72)	3.50	−0.44	+0.02	0.32

**Table 2 ijerph-20-02590-t002:** The Pearson *r* correlation between every time perspectives of the ZTPI-16 and each subscale of the ZTPI-56, ZTPI-36, ZTPI-30, or ZTPI-22. In bold are the significant correlations between two corresponding time perspectives of different versions of the ZTPI.

	PN-16	PP-16	PH-16	PF-16	F-16
**PN-56**	**+0.83** **	+0.08 **	+0.16 **	+0.34 **	+0.005
**PP-56**	−0.16 **	**+0.86** **	−0.02	−0.03	+0.26 **
**PH-56**	+0.08 **	+0.26 **	**+0.70** **	+0.21 **	−0.03
**PF-56**	+0.27 **	+0.09 **	+0.12 **	**+0.85** **	−0.07 **
**F-56**	+0.01	+0.21 **	−0.05 *	−0.22 **	**+0.80** **
**PN-36**	**+0.45** **	+0.33 **	+0.23 **	+0.42 **	+0.34 **
**PP-36**	+0.30 **	**+0.51** **	+0.19 *	+0.34 **	+0.35 **
**PH-36**	+0.37 **	+0.32 **	**+0.25** **	+0.29 **	+0.20 *
**PF-36**	+0.39 **	+0.26 **	−0.03	**+0.40** **	+0.44 **
**F-36**	+0.21 *	+0.33 **	+0.16 *	+0.17 *	**+0.59** **
**PN-30**	**+0.68** **	+0.09	+0.13 *	+0.23 **	+0.06
**PP-30**	+0.05	**+0.63** **	+0.05	+0.001	+0.31 **
**PH-30**	+0.11 *	+0.24 **	**+0.44** **	+0.15 *	+0.16 *
**PF-30**	+0.24 **	+0.06	+0.17 *	**+0.65** **	−0.04
**FP-30**	+0.09	+0.39 **	−0.06	−0.04	**+0.77** **
**FN-30**	+0.40 **	+0.16 *	+0.19 **	+0.50 **	+0.11 *
**PH-22**	+0.03	+0.03	**+0.40** **	−0.11 *	−0.15 **
**PF-22**	+0.14 **	+0.16 **	+0.10 *	**+0.46** **	−0.11 *
**F-22**	+0.02	+0.28 **	+0.01	−0.15 **	**+0.63** **

Note * *p* < 0.05 and ** *p* < 0.0001.

## Data Availability

The raw data supporting the discussion of this article will be made available by the authors, without undue reservation.

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
