# Peer review of "The Italian Validation of the Zimbardo Time Perspective Inventory and Its Comparison with Three Time Perspective Inventories"

_ijerph, 2023, doi:10.3390/ijerph20032590_

Round 1

Reviewer 1 Report

Title: The Italian Validation of Zimbardo Time Perspective Inventory and Its Comparison With Three Time Perspective Inventories

Journal: International Journal of Environmental Research and Public Health

This research examined the structural validity and internal reliability of both the regular (56 items) and short form (16 items) of the Italian Zimbardo Time Perspective Inventory. The organization of the manuscript is good and written in a scholarly manner. Although it has the potential to be published, it has some major flaws that need to be addressed before publication. Please see my specific comments below:

  1. The authors did not clearly explain some aspects of the design of the study. For example, it needs to be made clear who the participants are. Similarly, the sampling procedure was not described. On a similar note, it is not clear how the sample size is determined. Authors are advised to provide these details. 

a.               It is pretty confusing to read in the method section that "STPI" was used, but in the results and discussion sections, it was referred to as ZTPI-22. A large number of participants (n = 708) filled out this scale, but a smaller number of participants filled out the shorter versions of ZTPI (specifically, S-ZTPI and ZTPI-36). Why are there relatively fewer participants filling out the ZTPI's shorter forms (specifically, S-ZTPI-30 and ZTPI-36)? Authors are advised to provide more details to clarify this issue. Thus, the authors must address how they determined the sample size and why the rate of participation for the other scales was drastically different across different versions of the scale. If this research was part of a larger study and we are reading a report of a portion of its findings, this has to be clearly indicated. 

b.               Design-wise, it is difficult to understand whether some participants opted not to fill out the scales or researchers provided them with only some of them. If it is the former, the authors need to explain and interpret the attrition rate and how it affects their results. On a different note, it seems participants received the scales in a specific order, i.e., the order of the scales was not randomized. Some results are likely to derive from the fact that items across different scales share remarkable similarities. Thus, some of the results could be due to methods of artifact, as participants found themselves rating similar (maybe sometimes identical) items. This seems to be a major problem for this research and needs to be addressed in the paper as a major weakness. 

  1. I expected to see correlations of ZTPI-16 factors with other time-related constructs to assess the validity of this version of the scale. However, authors utilized various forms of ZTPI (22item, 30item, 36item, 56item) to evaluate just the concurrent validity of the ZTPI-16. The authors mentioned in the introduction section that these other forms of the scale have validity problems, and naturally, it applies to the shorter versions of the ZTPI. Thus, if these versions are problematic, it is not very informative to know that this new measure (ZTPI-16) correlated with them strongly. Also, the subscale correlations of ZTPI-16 with ZTPI-56 seem large, but the correlations with shorter forms were relatively low. This difference was not elaborated on in the discussion section. This issue brings the question: do these scales even measure the same construct? Without a reference to any other source (i.e., assessment of correlations between ZTPI subscales and other relevant constructs that displayed correlations with ZTPI factors in prior research), it is difficult to evaluate this version's (ZTPI-16) usefulness and construct validity. 
  2. The authors report the findings of CFA of ZTPI-16 on page 6 lines 258-260. Although the authors describe this result as a "good fit", it is below the criteria of a good model fit (also described in the paper on p.5, lines 216-219). Sure, it seems that the shorter form yields better fit indexes than the ZTPI-56. However, it still fails to attain the criterion of good model fit as accepted in the field. Authors are advised to describe this finding more accurately in the paper, as it was not described accurately in several places in the manuscript, specifically the results and discussion sections. Also, if the 16-item version of the ZTPI is not attaining good structural validity, how useful is it to use this scale over the other versions? Authors are advised to compare the CFA findings from different studies, including ZTPI-22, ZTPI-30, and ZTPI-36. What would have been ideal is to conduct CFA on these versions based on the data collected in this research; however, I suspect the sample sizes are a bit small for ZTPI-30 and ZTPI-36, though this could be done for ZTPI-22 where n = 708. This approach would provide more information to decide whether ZTPI-16 is superior to the other ZTPI versions in the Italian context. 
  3. Even if the changes suggested above are fixed, it is not clear how this study contributes to the measurement issues in the time perspective literature. The results and inferences seem very narrow as they apply only to the Italian context and may be irrelevant in other contexts. It is difficult to generalize current findings to cross-cultural studies of time perspective. Also, it does not decisively address and resolve measurement issues in the field. Alternatively, these contributions could be better specified and highlighted in the discussion section. 

Author Response

Dear Editor:

We are grateful to the editor, section managing editor, and reviewers for taking the time to carefully review the manuscript and give detailed and constructive comments. In the following part of this letter, we responded to all comments provided by reviewers and an English mother tongue revised the paper to improve the English language and style. All changes in the manuscript are in red.

REVIEWER#1

(x) English language and style are fine/minor spell check required

An English mother-tongue revised the paper to improve the English language and style. All changes are in red.

Yes

Can be improved

Must be improved

Not applicable

Does the introduction provide sufficient background and include all relevant references?

(x)

( )

( )

( )

Are all the cited references relevant to the research?

(x)

( )

( )

( )

Is the research design appropriate?

( )

( )

(x)

( )

Are the methods adequately described?

( )

( )

(x)

( )

Are the results clearly presented?

( )

(x)

( )

( )

Are the conclusions supported by the results?

( )

( )

(x)

( )

We thank the reviewer#1 for positive comments for introduction and references quoted. We also took in consideration the need to improve the methods, design and conclusions, according to the reviewer’s comments (see below).

Comments and Suggestions for Authors:

This research examined the structural validity and internal reliability of both the regular (56 items) and short form (16 items) of the Italian Zimbardo Time Perspective Inventory. The organization of the manuscript is good and written in a scholarly manner. Although it has the potential to be published, it has some major flaws that need to be addressed before publication”.

We thank the reviewer#1 for his/her positive comment for the manuscript and the potential to be published. In the following part of this letter, we replied to specific comments provided by the reviewer#1.

  1. The authors did not clearly explain some aspects of the design of the study. For example, it needs to be made clear who the participants are. Similarly, the sampling procedure was not described. On a similar note, it is not clear how the sample size is determined. Authors are advised to provide these details”.

We apologized whether the design of the study was not clear. As written in the participants section, all participants were volunteers, unpaid and anonymous. They were recruited in the university courses (medicine and psychology), whereas other participants were recruited through flyers and social media post. Thus, participants were selected on the basis of their willingness to participate as performed in previous studies (e.g., Orosz et a., 2017; Molinari et al., 2016, Laghi et al., 2013; Temple et al., 2019; Worrell et al., 2018; Davis & Ortiz, 2017; McKay et al., 2015; Worrell & Mello, 2007; Milfont et al., 2008; Wang et al., 2015; Orkibi, 2014; Akirmak, 2021; Skogen & Nesvåg, 2019). This sampling procedure seemed to proceed through opportunistic (e.g., convenience) and snowball sampling. As regards the sample size, we agreed with what Peng et al. (2021) wrote at page 3:“Determining sample size requirements for CFA remains a challenge, as the requirements are impacted by the number of factors and indicators, as well as the magnitude of factor loadings (Wolf et al., 2013, Educ.Psychol. Meas., 76, 913.934)”. Thus, we assessed the numerosity in all previous studies: Sircova et al. (2014) included 26 samples (covering studies from 2004 to 2012) from 24 countries for a total of 12.200 participants and the sample sizes ranged from 180 (United Kingdom) to 1.269 (Russia); Worrell et al. (2015) included 1.620 participants; Peng et al. (2021) recruited 1.393 participants whereas Košt’ál et al. (2016) recruited 2.062 individuals. In Italian culture, D’Alessio et al. (2003) assessed 1.507 individuals, Molinari et al. (2016) included 435 participants, in similar way to Carelli et al. (2011) with N = 417, and Laghi et al. (2013) reported 1.300 adolescents. Taking into account that the numerosity in the single studies ranged from 303 (Wang et al., 2015) to 1.370 (Orosz et al., 2017), while in the multicenter studies N varied from 1.150 (Sobol-Kwapinska et al., 2018) to 3.306 (Worrell et al., 2018), we could think that our numerosity (N = 2.295) was in line with the sample sizes of studies in the literature. However, we decided to add details about the sampling procedure and sample size (page 4 lines 153-183).

  1. It is pretty confusing to read in the method section that "STPI" was used, but in the results and discussion sections, it was referred to as ZTPI-22. A large number of participants (n = 708) filled out this scale, but a smaller number of participants filled out the shorter versions of ZTPI (specifically, S-ZTPI and ZTPI-36). Why are there relatively fewer participants filling out the ZTPI's shorter forms (specifically, S-ZTPI-30 and ZTPI-36)? Authors are advised to provide more details to clarify this issue. Thus, the authors must address how they determined the sample size and why the rate of participation for the other scales was drastically different across different versions of the scale. If this research was part of a larger study and we are reading a report of a portion of its findings, this has to be clearly indicated”.

We agreed with the reviewer#1 to uniform the name of the questionnaires in order to not create confusion. The research was structured to validate the ZTPI-16 in comparison to other versions of the ZTPI (ZTPI-56, ZTPI-22, ZTPI-36 and ZTPI-30) only. The first attempt was to assess the reliability and validity of the ZTPI-56 in Italian context given that no studies were available in the literature, and, thus, only the original ZTPI-56 was administered. In addition, this procedure was adopted in the previous studies in which a short and/or modified version of the ZTPI was tested (e.g., Peng et al., 2021). The ZTPI-16 was then extracted from the corresponding items of the ZTPI-56, as described in the original version of the manuscript in the Data Analysis section. The remaining sample received the ZTPI-56 and one of the ZTPI-22, ZTPI-30, and ZTPI-36. The rate of participation for the other scales was different as different was the citations for each article. Specifically, D’Alessio et al. (2003) received 358 citations with Goggle Scholar and 160 in Scopus, Sircova et al. (2014) received 174 citations with Goggle Scholar and 114 in Scopus, and, finally, Molinari et al. (2016) received 32 citations with Goggle Scholar and 8 in Scopus. Moreover, the aim of the paper was to assess the validity and reliability of the ZTPI-16 and, in our opinion, it was necessary to test this version with other short version (ZTPI-22), while the other two versions contained more items. Thus, these aspects determined the different numbers of participants with ZTPI-56 and ZTPI-22, ZTPI-30, or ZTPI-36. We decided to specify aspects of the procedure in the manuscript (page 4, lines 174-178).

  1. Design-wise, it is difficult to understand whether some participants opted not to fill out the scales or researchers provided them with only some of them. If it is the former, the authors need to explain and interpret the attrition rate and how it affects their results. On a different note, it seems participants received the scales in a specific order, i.e., the order of the scales was not randomized. Some results are likely to derive from the fact that items across different scales share remarkable similarities. Thus, some of the results could be due to methods of artifact, as participants found themselves rating similar (maybe sometimes identical) items. This seems to be a major problem for this research and needs to be addressed in the paper as a major weakness”.

As regards the design, we hoped that the previous response clarified this aspect. In the present research the researchers provided the scales and no participant’s option was provided. Thus, no explanation and interpretation of the attrition rate should be provided. We thank the reviewer#1 for noting the lack of specific reference to the order of scales administered to participants. We clarified this aspect (page 4, lines 179-180).

  1. I expected to see correlations of ZTPI-16 factors with other time-related constructs to assess the validity of this version of the scale. However, authors utilized various forms of ZTPI (22item, 30item, 36item, 56item) to evaluate just the concurrent validity of the ZTPI-16. The authors mentioned in the introduction section that these other forms of the scale have validity problems, and naturally, it applies to the shorter versions of the ZTPI. Thus, if these versions are problematic, it is not very informative to know that this new measure (ZTPI-16) correlated with them strongly. Also, the subscale correlations of ZTPI-16 with ZTPI-56 seem large, but the correlations with shorter forms were relatively low. This difference was not elaborated on in the discussion section. This issue brings the question: do these scales even measure the same construct? Without a reference to any other source (i.e., assessment of correlations between ZTPI subscales and other relevant constructs that displayed correlations with ZTPI factors in prior research), it is difficult to evaluate this version's (ZTPI-16) usefulness and construct validity”.

We can agree with the reviewer#1 that the correlation between the ZTPI-16 with other time-related constructs can give more emphasis to our paper. However, we follow the similar procedure adopted by Peng et al. (2021, for the ZTPI-16) and Košt’ál et al. (2016, for their ZTPI-15). Nevertheless, to the best of our knowledge, we reported moderate (.40)-to-strong (.80 and much more) r values claiming for good reliability of the ZTPI-16 with other versions of the ZTPI. In addition, the present study was the first attempt to replicate the study by Peng et al. (2021) in other culture. In this way, we wanted to be sure to test the reliability and validity of the ZTPI-16, given that it represents an attempt to provide a ZTPI with only good items, which has better psychometric properties. Thus, our procedure could be adequate to our aim, and we hoped that this study could be followed by subsequent studies to further assess the validity of the presented ZTPI-16. Although the ZTPI-22, ZTPI-30 and ZTPI-36 presented some criticisms, these tools have good psychometric properties. We decided to clarify this aspect in the introduction (page 3, lines 123-149). Consequently, we disagreed with the reviewer#1 for the question about the lack of information of strong correlations between ZTPI-16 and problematic versions of the ZTPI, questioning the construct of the different scales used in the present research. However, we recognized that the discussion needed a more elaboration of the difference found in the correlations between ZTPI-16 and ZTPI-56, from one hand, and between ZTPI-16 and ZTPI-30 or ZTPI-36, on the other hand (page 9 lines 404-440).

  1. The authors report the findings of CFA of ZTPI-16 on page 6 lines 258-260. Although the authors describe this result as a "good fit", it is below the criteria of a good model fit (also described in the paper on p.5, lines 216-219). Sure, it seems that the shorter form yields better fit indexes than the ZTPI-56. However, it still fails to attain the criterion of good model fit as accepted in the field. Authors are advised to describe this finding more accurately in the paper, as it was not described accurately in several places in the manuscript, specifically the results and discussion sections. Also, if the 16-item version of the ZTPI is not attaining good structural validity, how useful is it to use this scale over the other versions? Authors are advised to compare the CFA findings from different studies, including ZTPI-22, ZTPI-30, and ZTPI-36. What would have been ideal is to conduct CFA on these versions based on the data collected in this research; however, I suspect the sample sizes are a bit small for ZTPI-30 and ZTPI-36, though this could be done for ZTPI-22 where n = 708. This approach would provide more information to decide whether ZTPI-16 is superior to the other ZTPI versions in the Italian context”.

The reviewer#1 is right in pointing out a more carefulness in interpretating the results given that the fit indexes of our ZTPI-16 are good but some of them did not reach the cut-off criteria for goodness. We modified this aspect in the text and especially in the discussion section. Then, we compared our CFA findings with those reported in the literature and tried to give a possible explanation regarding the useful of this version, although not all indexes reached the criterion of good model fit (page 9, lines 350-372). However, we also agreed with the position expressed by Perry (the reviewer#2) et al. (see reference number [67] in the reference section of the revised paper) who suggested to not be strictly adherent to these cut-off values especially in social sciences. And thus, we think that our short version seems to be a good solution to the measurement issues in the time perspective literature and in Italian context. Finally, the goodness of this version was further confirmed by running additional suggested data analyses provided by the reviewer#2.

  1. Even if the changes suggested above are fixed, it is not clear how this study contributes to the measurement issues in the time perspective literature. The results and inferences seem very narrow as they apply only to the Italian context and may be irrelevant in other contexts. It is difficult to generalize current findings to cross-cultural studies of time perspective. Also, it does not decisively address and resolve measurement issues in the field. Alternatively, these contributions could be better specified and highlighted in the discussion section”.

We are sorry that the reviewer#1 did not appreciate our attempt to address measurement issue in the field. In our opinion, the substantial replication of the results reported by Peng et al. (2021) can represent a proof that, probably, our findings could be generalize in different cultures. Although we focused on Italian context, the similarity of the findings with Peng et al. in the Chinese context could be advance the idea that the ZTPI-16 is a potential attempt to address (and resolve) measurement issue in the time perspective field, irrespectively from specific cultural situations. In the original version of the paper we already tried to give insight for the contribution of this paper in the field. It is important, in our opinion, to bear in mind that the ZTPI is the most widely used tool in time perspective literature and an attempt to resolve measurement issue is relevant. However, we tried to highlight the contribution of this manuscript in the introduction (page 3, lines 116-122) and discussion sections (page 9, lines 377-385).

Reviewer 2 Report

This manuscript details a study testing the factor structure of several short versions of the ZTPI in a large sample of Italian adults. A 16-item version is of particular focus and is presented as a valid a reliable way to move forward. There are some clear strengths to the study, such as the justification for the items contained in the 16-item version, and the sample size. I have serious concerns over whether this manuscript yet adds clarity to the problem, however. I present any criticisms here in a collegial way that I hope is constructive. I would dearly like to see this area of research gain greater clarity and therefore, my suggestions are purely with this aim in mind, and I wish the authors well.

I don’t fully agree with the assertion on page 2, lines 71-74 that “ All these solutions have shown acceptable psychometric properties in terms of both reliability and validity, suggesting that it is possible to identify those items which can in a better way resolve the psychometric problems of the original 56-items scale.” Firstly, not all of these solutions have provided acceptable psychometric properties. The shortest measures often yield very weak internal consistency. I think it is prudent in this area to not make strong statements about a ZTPI version based on a single sample. Where the solutions cited have presented acceptable psychometric properties, these have often not been reproduced in other samples.

In the following paragraph, which is largely very good, I don’t necessarily agree with the concluding statement that the CFA results in Worrell et al., question the utility of a theory-driven approach. I think they question the utility of using the existing items. What the ZTPI needs, is re-writing.

I don’t believe that a CFA, independent clusters model is the most appropriate approach to examine multidimensional scales. This incorrectly notes non-significant cross-loadings as misspecifications and therefore, only supports models where factors are completely independent. This is untrue in most trait-based multidimensional operationalisations. See doi.org/10.1080/1091367X.2014.952370

Peng et al. treated the data as ordinal, using the DWLS method. I presume the data here has been subjected to the default, maximum likelihood method? This detail is required. I have yet to really form an opinion on the appropriateness of Peng et al.’s approach, but it treats ZTPI data very differently from all the other studies. If their results are attempted to be replicated, this is an important consideration.

The results need more detail. For example, measurement invariance should be tested. Methods other than the crude CFA-ICM are required. Internal consistency estimates are important – not just IIC.

In summary, I think this is a useful examination of Peng’s model, but would like to see some more testing within this sample. Given the size of the sample, I wonder if it could be randomly divided in half and tested on both samples to see if there is any variation in model it. Again, this could be done by testing measurement invariance. I’m not sure which software the authors use. I use MPlus, but I am happy to send on any syntax for testing ZTPI models (CFA, ESEM etc. and showing Peng’s method) and testing invariance.

Author Response

Dear Editor:

We are grateful to the editor, section managing editor, and reviewers for taking the time to carefully review the manuscript and give detailed and constructive comments. In the following part of this letter, we responded to all comments provided by reviewers and an English mother tongue revised the paper to improve the English language and style. All changes in the manuscript are in red.

REVIEWER#2

Open Review

(x) English language and style are fine/minor spell check required

An English mother-tongue revised the paper to improve the English language and style. All changes are in red.

Yes

Can be improved

Must be improved

Not applicable

Does the introduction provide sufficient background and include all relevant references?

( )

(x)

( )

( )

Are all the cited references relevant to the research?

(x)

( )

( )

( )

Is the research design appropriate?

( )

(x)

( )

( )

Are the methods adequately described?

( )

(x)

( )

( )

Are the results clearly presented?

( )

( )

(x)

( )

Are the conclusions supported by the results?

( )

(x)

( )

( )

We thank the reviewer#2 for his positive evaluations of the paper about the cited references. We also took in considerations that introduction, research design, method, results, and conclusions should be improved, according to reviewer’s comments (see below).

Comments and Suggestions for Authors

This manuscript details a study testing the factor structure of several short versions of the ZTPI in a large sample of Italian adults. A 16-item version is of particular focus and is presented as a valid a reliable way to move forward. There are some clear strengths to the study, such as the justification for the items contained in the 16-item version, and the sample size”.

We thank the reviewer#2 for the positive evaluations, highlighting the strengths of the paper, such as the justification and the sample size.

I have serious concerns over whether this manuscript yet adds clarity to the problem, however. I present any criticisms here in a collegial way that I hope is constructive. I would dearly like to see this area of research gain greater clarity and therefore, my suggestions are purely with this aim in mind, and I wish the authors well”.

We are sorry whether the reviewer#2 has concerns about the clarity of the manuscript regarding the measurement problem in the field. Our paper was an attempt to investigate the appropriateness of Peng et al.’ approach, and, in some way, our data seemed to suggest that this approach deserves further attention and investigation in subsequent studies. We really appreciate your comments and criticisms because they gave us impulses to improve the quality of the manuscript in order to contribute to the field.

I don’t fully agree with the assertion on page 2, lines 71-74 that “All these solutions have shown acceptable psychometric properties in terms of both reliability and validity, suggesting that it is possible to identify those items which can in a better way resolve the psychometric problems of the original 56-items scale.” Firstly, not all of these solutions have provided acceptable psychometric properties. The shortest measures often yield very weak internal consistency. I think it is prudent in this area to not make strong statements about a ZTPI version based on a single sample. Where the solutions cited have presented acceptable psychometric properties, these have often not been reproduced in other samples”.

The reviewer#2 is right, and we reduced the strength of the statements at the end of this paragraph (page 2, lines 72-75). The original meaning of these statements was to introduce the possibility that a short version of the ZTPI could address the measurement problems reported for the ZTPI-56 (without abandoning this questionnaire given that it is the most widely used tool in the field). This part of the introduction, in our opinion, is/was necessary to justify our choice to test the psychometric properties of a 16-items version. In addition, this part is useful to introduce the subsequent paragraph in order to introduce in better way the rationale of the study.

In the following paragraph, which is largely very good, I don’t necessarily agree with the concluding statement that the CFA results in Worrell et al., question the utility of a theory-driven approach. I think they question the utility of using the existing items. What the ZTPI needs, is re-writing”.

Of course, we agreed with the reviewer#2 and thank him for this suggestion. As co-author of the paper “A theoretical approach to resolving the psychometric problems associated with the Zimbardo Time Perspective Inventory”, the reviewer#2 gave the correct message of the above study. We modified this part accordingly (pages 2-3, lines 93-99).

I don’t believe that a CFA, independent clusters model is the most appropriate approach to examine multidimensional scales. This incorrectly notes non-significant cross-loadings as misspecifications and therefore, only supports models where factors are completely independent. This is untrue in most trait-based multidimensional operationalisations. See doi.org/10.1080/1091367X.2014.952370.”

We would like to thank very much the reviewer for his comments and his precious help, because we believe that his help significantly has strengthened the data analysis of our paper. In the revised version, the ESEM analyses were presented (page 5, lines 237-249). The fit indexes were in line with those observed in the previous CFA, but factors resulted to be more related, as the reviewer#2 anticipated. Results and Data analysis paragraphs were changed to describe new ESEM analyses, and related bibliographic references were added (pages 6, lines 279-299, and see references [62] and [63]).

Peng et al. treated the data as ordinal, using the DWLS method. I presume the data here has been subjected to the default, maximum likelihood method? This detail is required. I have yet to really form an opinion on the appropriateness of Peng et al.’s approach, but it treats ZTPI data very differently from all the other studies. If their results are attempted to be replicated, this is an important consideration”.

We apologized that in the original version of the paper we did not include any specification of the estimates used. As the reviewer#2 pointed out, Peng et al. treated ZTPI data very differently from the other studies. In our analyses we opted for the maximum likelihood estimation with robust standard errors, that is used to protect from deviation of normality in the indicator variables. In the new version of the paper, the use of MLR estimation was added in the data-analysis paragraph (page 5, lines 240-242).

The results need more detail. For example, measurement invariance should be tested. Methods other than the crude CFA-ICM are required. Internal consistency estimates are important – not just IIC”.

We hoped that in this revised paper the reviewer#2 could be satisfied with the result section, which, in our opinion, was more detailed (pages 6-7).

In summary, I think this is a useful examination of Peng’s model, but would like to see some more testing within this sample. Given the size of the sample, I wonder if it could be randomly divided in half and tested on both samples to see if there is any variation in model it. Again, this could be done by testing measurement invariance. I’m not sure which software the authors use. I use MPlus, but I am happy to send on any syntax for testing ZTPI models (CFA, ESEM etc. and showing Peng’s method) and testing invariance. If this would be helpful, drop me an email on John.L.Perry@ul.ie”.

First of all, we would like to thank the reviewer#2 for sharing with us his syntax to test ZTPI models. In this revised version of the paper, following the reviewer#2’s  comment, we conducted a multigroup analysis and we randomly divided in half the sample and tested on both samples the configural, metric and scalar invariance of the scale (page 5, lines 244-249).